# Contribution of Orbital Ultrasound to the Diagnosis of Central Retinal Artery Occlusion

**DOI:** 10.3390/jcm11061615

**Published:** 2022-03-15

**Authors:** Laura Rojas-Bartolomé, Óscar Ayo-Martín, Jorge García-García, Francisco Hernández-Fernández, Elena Palazón-García, Tomás Segura

**Affiliations:** 1Department of Neurology, Complejo Hospitalario Universitario de Albacete, c/Hermanos Falco 37, 02006 Albacete, Spain; oscarayo@gmail.com (Ó.A.-M.); jggpillarno@hotmail.com (J.G.-G.); fco.hdez.fdez@gmail.com (F.H.-F.); elena83pg@hotmail.com (E.P.-G.); tseguram@gmail.com (T.S.); 2Instituto de Investigación en Discapacidades Neurológicas (IDINE), Facultad de Medicina, Universidad de Castilla-La Mancha, 02008 Albacete, Spain

**Keywords:** orbital ultrasound, central retinal artery occlusion, arterial occlusive diseases, retinal spot sign, embolism

## Abstract

We aimed to evaluate the diagnostic value of orbital ultrasound in the etiologic diagnosis of central retinal artery occlusion (CRAO). For this purpose, patients with CRAO evaluated at our center between 2011 and 2021 were reviewed. Demographic variables, vascular risk factors and ultrasound findings were collected. An orbital duplex was performed in all cases and complemented with other diagnostic explorations. We attended 36 cases of CRAO. In all patients, orbital ultrasound confirmed the diagnosis of CRAO: in 75% emboli material (spot sign) was observed in CRA and in 25% flow alteration in CRA without visible embolus. The positive spot sign (PSS) group differed from patients with negative spot sign (NSS) in terms of etiology: 8 PSS cases (29.6%) had a major cardioembolic cause, 4 (14.8%) a large vessel atheromatous disease, 15 (55.6%) an undetermined cause. Some 21 (77.8%) PSS patients had some minor cardioembolic cause, mainly calcifications of the left valves. In the NSS group, 2 (22%) were diagnosed with giant cell arteritis (GCA). In CRAO, the ultrasound spot sign could be a guide for the detection of embolic sources. Its absence makes it necessary to consider more strongly the possibility of arteritis. Furthermore, our findings suggest a key role of calcium embolism in PSS patients.

## 1. Introduction

Central retinal artery occlusion (CRAO) is a medical emergency that causes sudden, painless and severe visual loss. In most cases, it is accompanied by an ominous prognosis since about 80% of patients will develop amaurosis with no subsequent improvement in visual acuity (VA < 20/400) [1].

The etiology of CRAO can vary widely [2,3,4,5], but most cases are secondary to migration of embolic material, either arterio-arterial (from a carotid atherosclerotic plaque or from the aortic arch) or of cardiac origin. As for cases of non-embolic origin, their hemodynamic cause has been postulated, either due to occlusion of the ipsilateral internal carotid artery (ICA), due to atheromatous disease of the ophthalmic artery or to inflammation and arteritic occlusion of the common trunk of the medial posterior ciliary artery and the central retinal artery (CRA), in the context of giant cell arteritis (GCA) [4,6,7].

Beyond the specific cause of CRAO, there is also evidence that retinal vascular pathology is closely linked to cerebrovascular disease. The presence of retinal embolisms has been described as a risk factor for stroke and mortality from cardiovascular disease [8] since a high risk of long-term stroke recurrence after an episode of CRAO has been detected [9]. In addition, 24% of patients with an acute retinal ischemic event have a concomitant cerebral infarction on MRI [10]. Thus, a CRAO event could be assumed to be a predictor of stroke, and given its analogous nature to acute ischemic stroke, CRAO should likely be treated as a “retinal stroke” [11].

The latest AHA guidelines recommend an ophthalmological evaluation to confirm the diagnosis of CRAO and rule out other disorders and suggest complementary tests such as optical coherence tomography, angiography, or fluorescein angiography [12]. However, an ophthalmologist is not always available in the Emergency Room on-site for clinical evaluation.

In this context, ocular ultrasound allows simple, reliable and rapid detection of several signs which suggest the presence and mechanism of a CRAO: the absence of flow in the CRA by color and pulsed Doppler, and the presence of hyperechoic material in the retrobulbar circulation of the optic nerve in B-mode. The last finding is called the spot sign, which has been suggested as a marker of an embolic mechanism at this level [13,14]. Despite its advantages, the use of orbital ultrasound is not common in the study of CRAO, and it is usually recommended only when there is a high suspicion of GCA [14,15].

The aim of our work is to show the usefulness of orbital ultrasound in the diagnosis of CRAO and to explore its potential value as a guide for the etiological study of this pathology based on the presence or absence of the spot sign.

## 2. Materials and Methods

A series of retrospective cases of CRAO evaluated at our center and collected in a database between the years 2011 and 2021 were reviewed. The initial diagnosis was made by the Ophthalmology Service through anamnesis and typical retinal findings in funduscopy (cherry-red spot, emboli, optic disk oedema, retinal pallor and arteriolar narrowing). In all cases, after the ophthalmologic diagnosis, an orbital ultrasound study was performed within 24 h by the Neurology Department.

Depending on the initial result of the orbital ultrasonography, we distinguished two large groups, PSS (positive spot sign) and NSS (negative spot sign with an absence of flow in the CRA). The etiological study in each patient was complemented with additional explorations at the discretion of the clinicians in charge.

Demographic variables and vascular risk factors were collected. Orbital ultrasound findings were obtained using a 3–9 MHz linear probe (Esaote MyLab70 and My Lab9, Esaote, Milan): the absence of color and pulsed Doppler signal in the peripapillary portion of the CRA (diagnosis of CRAO) (Figure 1) and the presence of hyperechoic material in the retrobulbar circulation of the optic nerve (spot sign) were explored (Figure 2).

The complementary ultrasound study was performed by duplex of the cervical and intracranial carotid system and of the anterior and posterior intracranial circulation (linear 3–9 MHz and sectorial 1–4 MHz probes, respectively) (Esaote MyLab70 and My Lab9, Esaote, Milan). In cases where there was clinical suspicion of GCA [16,17], ultrasound study of temporal and occipital arteries [18] was performed (sectorial probe 6–18 MHz) (Esaote MyLab70, Esaote, Milan). Following the official guidelines for the diagnosis of ethiology in stroke, according to the previous findings at the discretion of the attending clinician, the etiological study was completed in some cases by EKG telemetry in the stroke unit, ambulatory EKG-Holter-24 h, transthoracic echocardiography or determination of right-to-left cardiac shunt by transcranial ultrasonography.

Because of the absence of consensus for the etiological classification of arterial pathology in the retinal territory, the TOAST classification for cerebral infarctions was used [19].

A descriptive statistical analysis was performed using the IBM SPSS Statistics 22 program. To test the association between categorical variables, the Chi-square test was used if the expected frequency value was less than 5 in less than 20% of the associations; otherwise, Fisher’s exact test was used. Statistical significance was defined in cases of *p* < 0.05.

## 3. Results

During the study period, 36 cases with a clinical diagnosis of CRAO were attended.

Orbital ultrasound confirmed the diagnosis of CRAO in all cases. Some 27 (75%) were classified as PSS and 9 (25%) as NSS. In the PSS group, the funduscopy study only detected embolic material in peripheral retinal branches in four patients, so that in more than 85% of cases in this group this finding had not been detected in the initial ophthalmological examination.

The mean age was 72.4 ± 12.01 (mean ± SD), similar in both groups. There were 38.9% women. The prevalence of arterial hypertension (AH), hyperlipidaemia (HLP), diabetes mellitus (DM), ischemic heart disease and peripheral atheromatous artery disease was higher in the PSS group. Smoking was similar in both groups. Nine (25%) patients had atrial fibrillation (AF) (eight in the PSS group and one in the NSS group), there were three cases (8.3%) of heart valve prosthesis (two cases in the PSS group and one in the NSS group) and one (2.7%) of endocarditis (PSS group) (Table 1).

A cervical carotid study was conducted in all patients: ultrasound study was performed in 35 (97.2%) patients, angio-CT in seven. Intracranial vascular evaluation (intracranial duplex and/or angio-CT of the circle of Willis) in 30 (83.3%) of them, and temporal and occipital arteries were explored in 14 (39%); transthoracic or transesophageal echocardiography was performed in 29 (81%, all of them without a patent cause of retinal ischemia shown by the previous tests) cases and EKG monitoring in 20 (55.6%, all of them without a patent cause of retinal ischemia shown by the previous tests), of these, 16 (44.4%) were admitted to the stroke unit, and four (11.1%) had ambulatory Holter monitoring. Table 2 shows the number of studies performed in each group. Of the seven patients in whom echocardiography was not performed, six already had a clear potential etiology for CRAO (4 patients had an ipsilateral significant ICA stenosis (≥50%) or occlusion, 1 had AF, in 1 case a diagnosis of GCA was made) and one patient was referred to another hospital before the study could be completed.

According to the TOAST classification (Table 3), nine of the 36 cases of CRAO (25%) were cardioembolic in origin, eight (22.2%) had large vessel atheromatous disease (three due to ICA occlusion and five due to ipsilateral significant ICA stenosis ≥ 50%, at the intracranial or cervical level), two (5.6%) were due to GCA and 17 (47.2%) were of undetermined origin.

There were clear differences in the etiology, according to the presence or absence or spot sign. In the PSS group, most cases with a determined cause were of cardioembolic etiology (29.6%), whereas in patients in the NSS group, the main etiology was an atheromatous disease of large vessels (44.4%) (Table 3). There was a high percentage of cases of undetermined origin in both groups.

Of note, there was a high proportion of minor cardioembolic causes in PSS patients (77.8%) whilst only 11% in NSS patients (Table 4). Noteworthy, all of the TOAST-undetermined patients (15 cases) in the PSS group had a minor cardioembolic source, but none of the TOAST-undetermined NSS cases. Valve calcifications were the main minor cardioembolic source, present in 70,4% of PSS, in 13 (86.7%) patients of undetermined cause and five (71.4%) with AF.

In the bivariate analysis, the PSS group presented statistically significant differences with the NSS group in the etiology (X2 (3) = 11.18, *p* = 0.011) according to the TOAST stroke classification.

Grouping all cardioembolic sources (major and minor) as a single cause, we also found statistically significant differences in etiology between the PSS and NSS groups (X2 (3) = 20.22, *p* = 0.000): atheromatous etiology 14.8% vs. 44.4%, cardioembolic 85.2% vs. 11.1%, GCA 0% vs. 22.2% and undetermined 0% vs. 22.2%, respectively. Furthermore, the PSS group was significantly associated with the presence of a unified cardioembolic etiology (X2 (1) = 16.66, *p* = 0.000). Five (71.42%) of the seven cases of major cardioembolic etiology (AF) in PSS group also had a minor cardioembolic source (left heart valves calcifications).

Finally, global atheromatous carotid disease (significant stenosis or not) was detected in 28 cases (77.8%) of the total, 22 (81.5%) in the PSS group and six (66.7%) in the NSS group. Nevertheless, ipsilateral significant ICA stenosis was more frequent in the NSS group (55.6%) versus the PSS (18.5%) significantly (X2 (2) = 8.47, *p* = 0.014); moreover, the absence of the spot sign was associated with the presence of GCA significantly (X2 (1) = 6.35, *p* = 0.012).

## 4. Discussion

Our series shows that orbital ultrasound can confirm CRAO by detecting the absence of flow within the CRA in all cases, whether or not accompanied by the detection of the spot sign. In our sample, there was a high percentage of cases in the PSS group without fundoscopic evidence of embolic material at the initial ophthalmologic examination, which indicates greater sensitivity of ultrasonography in demonstrating the embolic nature of CRAO and makes it potentially more useful in guiding the clinician. It is noteworthy that these data replicate the findings described in the previous studies [20].

In addition, orbital ultrasound in the acute phase, by detecting or not the spot sign, could help to guide the etiological study, crucial in certain cases to schedule secondary prevention treatment in routine clinical practice. Our study shows that there were clear differences between the two groups (PSS and NSS) in terms of etiology according to the TOAST classification of stroke. The PSS group had most cases of major cardioembolic etiology (29.6%), and, in general, 85% of the PSS cases had some type of cardioembolic source (major or minor according to the TOAST classification). On the other hand, in the NSS patients, the variety of etiologies was much wider, and although the main etiology found was atheromatous large-vessel disease (44.4%), it is in the NSS group where all the cases of GCA were found.

Then, the difference found between the etiologies of the two groups could suggest a difference in the mechanism of occlusion in the CRA.

Likewise, as mentioned above, the only two cases of GCA were diagnosed in the NSS group, and it was carried out thanks to the ultrasound study of temporal and occipital arteries [18]. Again, only in the NSS group did we find cases of ipsilateral ICA occlusion. In our series, there were only four potential cases: an ICA occlusion due to cardiac embolism and three ICA occlusions due to large vessel atheromatous disease. We postulate a hemodynamic mechanism of retinal ischemia for these cases, given that we found no flow signal in the CRA in all the four patients (in two of them, there was even no flow in the ophthalmic artery). This hemodynamic mechanism of ischemia would be similar to that accepted for the so-called “ocular ischemic syndrome”, where high-grade stenosis or occlusion of the ICA, associated with poor collaterals of the external carotid branches, causes ischemia in the retina with presumed hemodynamic pathophysiology.

Our results may also help to clarify what causes CRAO in patients classified as “of undetermined cause”. In previous studies, there has been much heterogeneity in the way of considering potential causes of retinal ischemia [3,6,10]. We decided to apply the TOAST criteria, similar to what is used for stroke, and have detected a high percentage of cardioembolic sources, whether major or minor, especially in the PSS group. This leads us to hypothesize that these sources are possibly related to the appearance of the spot sign. More specifically, we were able to show that 70.4% of the patients in the PSS group had calcifications in the left valves (71.42% in patients with associated AF and 70% in patients without AF). In fact, although there are no histological studies that confirm the components of the spot sign, several authors favor its calcium or cholesterol composition, especially due to its persistence over time and the resistance shown by these emboli to treatment with thrombolytics [20,21]. Thus, in some series [22] and isolated cases [23,24], calcium microemboli have been described as a plausible cause of retinal ischemia, and retinal calcific emboli are proposed as the first clinical manifestation of cardiac valvular pathology in patients with calcific valvular stenosis [25]. In our series, the presence in the PSS group of a high proportion of calcium and cholesterol sources (calcified heart valves and/or carotid atheromatosis) could support the theory that the image of the spot sign is formed by either of these two substances.

Among the limitations of our study, we highlight the disparity in the performance of complementary tests in both groups. The etiological study was guided by the clinical history and the results of the initial ocular and cervical ultrasound; however, in seven cases, it was not completed. One patient was transferred of his own accord to his referral center, and in the other six cases, no cardiac (five cases) or intracranial vascular (one case) monitoring study was performed. None of these seven patients required hospital admission, which in our opinion, increases the probability of etiological study being incomplete. We understand that this decreases the sensitivity of the study performed with an excess of cases of undetermined cause.

## 5. Conclusions

Based on our results, we consider that the finding of the spot sign should act as a guide to an exhaustive search for an embolic source, which implies a correct evaluation of the carotid and cardiac system in each case of CRAO. It also made it possible to detect unusual embolic sources such as endocarditis. We also consider relevant the detection of findings considered “minor” such as cardiac valvular calcifications, due to their possible etiopathogenic role in cases considered of undetermined nature, and possibly also in some cases with major cardioembolic sources. On the other hand, in the absence of the spot sign, the search for symptoms or signs of GCA should be initiated early and obligatorily before continuing with the cardiological and vascular study, given the devastating nature of this pathology.

In conclusion, the inclusion of orbital ultrasound could improve the sensibility of early diagnosis of CRAO. It also can be crucial in clarifying the etiological cause of the event according to the presence or absence of the spot sign.

## Figures and Tables

**Figure 1 jcm-11-01615-f001:**
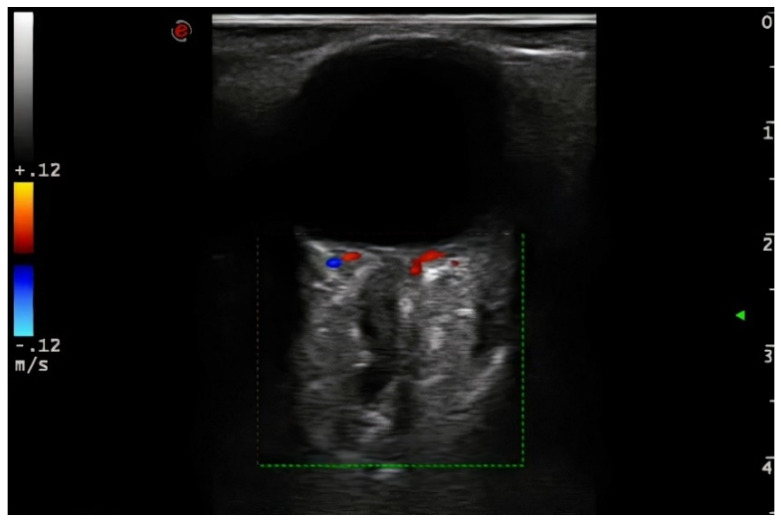
The absence of color and pulsed Doppler signal in the peripapillary portion of the CRA (diagnosis of CRAO). 3–9 MHz linear probe (Esaote MyLab70 and My Lab9, Esaote, Milan).

**Figure 2 jcm-11-01615-f002:**
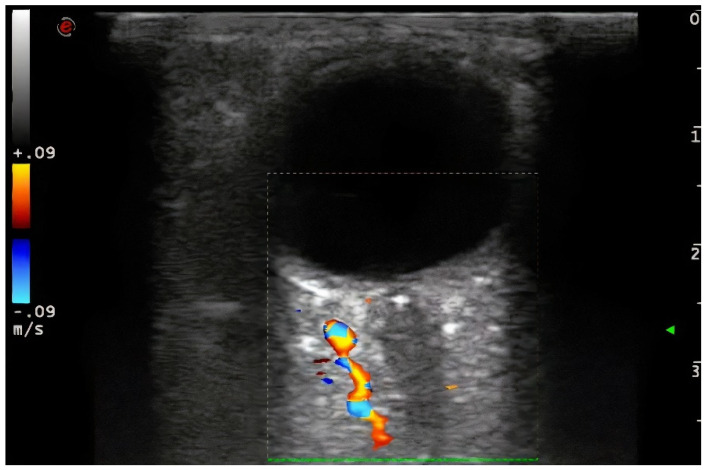
Hyperechoic material in the retrobulbar circulation of the optic nerve (spot sign). Note the abscense of color code flow signal in the peripapillar segment of the CRA. 3–9 MHz linear probe (Esaote MyLab70 and My Lab9, Esaote, Milan).

**Table 1 jcm-11-01615-t001:** Demographic characteristics and vascular risk factors.

	All Patients		PSS	NSS	*p* Value
Sample size, no. (%)	36		27 (75)	9 (25)	
Women, no (%)	14 (38.9)		10 (37)	4 (44.4)	0.9
Age (mean ± SD)	72.44 ± 12.01		72.07 ± 11.84	73.56 ± 13.2	0.75
Arterial hypertension, no. (%)	29 (80.6)		23 (85.2)	6 (66.7)	0.22
Hyperlipidaemia, no. (%)	20 (55.6)		16 (59.3)	4 (44.4)	0.44
Diabetes mellitus, no. (%)	8 (22.2)		7 (25.9)	1 (11.1)	0.35
Smoking, no. (%)	Smoker	3 (8.3)	2 (7.4)	1 (11.1)	0.68
	Ex-smoker	10 (27.8)	8 (29.6)	2 (22.2)	
Ischemic heart disease, no. (%)	7 (19.4)		6 (22.2)	1 (11.1)	0.46
Atheromatous artery disease, no. (%)	4 (11.1)		4 (14.8)	0	0.27
Atrial fibrillation, no. (%)	9 (25)		8 (29.6)	1 (11.1)	0.26
Heart valve prosthesis, no. (%)	3 (8.3)		2(7.4)	1 (11.1)	0.72

**Table 2 jcm-11-01615-t002:** Complementary tests.

Test	All Patients	PSS	NSS	*p* Value
Cervical carotid ultrasound, no. (%)	35 (97.2)	26 (96.3)	9 (100)	0.56
Orbital ultrasound, no. (%)	36 (100)	27 (100)	9 (100)	-
Intracranial duplex, no. (%)	30 (83.3)	21 (77.7)	9 (100)	0.07
Temporal and occipital artery duplex, no (%)	14 (39)	12 (44.4)	2 (22.2)	0.23
Angio-CT of the circle of Willis, no. (%)	7 (19.4)	4 (14.8)	3 (33.3)	0.16
Right-to-left cardiac shunt, no. (%)	2 (6)	2 (7.4)	0	0.4
Echocardiography, no. (%)	29 (81)	24 (88.9)	5 (55.6)	0.03
EKG monitoring, no. (%)	20 (55.6)	17 (63)	3 (33.3)	0.12
Admitted to the stroke unit, no. (%)	16 (44.4)	13 (48.1)	3 (33.3)	0.43
Ambulatory Holter monitoring, no. (%)	4 (11.1)	4 (14.8)	0	0.16

**Table 3 jcm-11-01615-t003:** CRAO etiology by TOAST classification.

TOAST Etiology	PSS (*n* = 27)	NSS (*n* = 9)	Total (*n* = 36) (No., % of Total)	*p* Value
Undetermined, no. (%)	15 (55.6)	2 (22.2)	17 (47.2)	0.083
Cardioembolic, no. (%)	8 (29.6)	1 (11.1)	9 (25)	0.267
Atrial fibrillation	7	1 ^‡^	8 (22.2)	
Mechanical heart valve prosthesis	1 ^†^	0	1 ^†^ (2.7)	
Marantic endocarditis	1	0	1 (2.7)	
Atheromatous large-vessel disease, no. (%)	4 (14.8)	4 (44.4)	8 (22.2)	0.22
Ipsilateral ICA stenosis occlusion	0	3	3 (8.3)	
Ipsilateral significant ICA stenosis (≥50%) cervical or supraclinoid	4	1	5 (13.8)	
GCA, no. (%)	0	2 (22.2)	2 (5.6)	0.012

^†^ Case with heart valve prosthesis and atrial fibrillation. ^‡^ Ipsilateral ICA stenosis occlusion due to cardiac embolism. Giant cell arteritis (GCA); internal carotid artery (ICA).

**Table 4 jcm-11-01615-t004:** Minor cardioembolic sources.

Minor Cardioembolic Sources	PSS (*n* = 27)	NSS (*n* = 9)	Total (*n* = 36)	*p* Value
Biologic heart valve prosthesis	1 (3.7%)	1 (11.1%)	2 (5.6%)	0.4
Left heart valves calcifications	19 (70.4%)	1 (11.1%) ^†^	20 (55.6%)	0.002
Right-to-left cardiac shunt	1 (3.7%)	0	1 (2.8%)	0.55
All minor cardioembolic sources	21 (77.8%)	1 (11.1%) ^†^	22 (61.1%)	0

^†^ Case of the NSS group with heart valve prosthesis and left valvular calcifications (atheromatous etiology). Positive spot sign group (PSS); Negative spot sign group (NSS).

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
