# Peer review of "Contribution of Orbital Ultrasound to the Diagnosis of Central Retinal Artery Occlusion"

_jcm, 2022, doi:10.3390/jcm11061615_

Round 1

Reviewer 1 Report

The authors described that the ultrasound spot sign could be a guide for the detection of embolic sources, and its absence would indicate atheromatous large-vessel pathology and GCA. The manuscript you wrote is based on the results of an analysis based on a detailed analysis. The above contents are considered to be useful results for future clinical practice.

I point out the following points to improve the accuracy of this manuscript.

・page1. Line13: embolic material was observed in CRA (spot sign) →emboli material (spot sign) was observed in CRA (central retinal artery)

・table 1 and 2: n°→no

・In all tables, the dots that indicate the numbers after the decimal point are commas.

Author Response

Response to Reviewer 1 Comments

We appreciate all the reviewer's comments and suggestions.  Attached is a point-by-point response to the reviewer's comments, as well as a copy of our revised manuscript based on the response to those comments. We will be happy to provide additional information upon request or to modify the text further if still necessary.

Point 1: page1. Line13: embolic material was observed in CRA (spot sign) →emboli material (spot sign) was observed in CRA (central retinal artery)

Response 1:  see comment “Point 1. Response to Reviewer 1”. I have changed the previous line to: “emboli material (spot sign) was observed in CRA (central retinal artery) “.

Point 2: table 1 and 2: n°→no

Response 2: see comment “Point 2. Response to Reviewer 1”. I have changed nº to no.

Point 3: In all tables, the dots that indicate the numbers after the decimal point are commas.

Response 3: see comment “Point 3. Response to Reviewer 1”. I have changed dots to commas.

We hope that these changes will provide a satisfactory response to the reviewer's comments.

We look forward to hearing from you soon. Thank you for your time and consideration.

Reviewer 2 Report

This study evaluated orbital ultrasound performed within 24h after ophthalmologically proven CRAO. An important finding is the confirmation of the ophthalmological diagnosis in all cases, which highlights the diagnostic strength of ultrasound in CRAO, thus reproducing literature data (Nedelmann et al Stroke 2015).

The results and discussion section is difficult to read and needs revision. 

There is a major issue that remains undiscussed. It is towards the nature of the spot sign. We do not know exactly its composition. But all circumstances are towards calcified or cholesterol particles and against pure blood clot: its bright reflex; it never disappears in follow-up or after thrombolysis: even after successful thrombolysis, it remains. That is clear evidence against atrial fibrillation as embolic source in SSP CRAO. This should be discussed more thoroughly. Respective literature should be cited and discussed (for example Altmann et al, J Neuroimaging 2015; Nedelmann et al Stroke 2015). In the context of this assumption, the finding of a strong association of heart valve calcification and SSP CRAO is the most interesting aspect of this study. It may well explain, why spot signs never disappear in follow-up (in contrast to most embolic spot sign negative CRAO that eventually recanalise spontaneously. To my knowledge, this finding has not been described previously.

 Line 119: A stenosis >/= 50% is not high grade, which would be >/= 70%. This should be corrected. Please state, how many of your patients with ICA stenosis had a retrograde ophthalmic artery (a retrograde flow would effectively preclude embolism from stenosis to CRA.

Line 147 and Table 5: I completely do not understand the assumption of hemodynamic CRAO in the stated context. Visual compromise due to low retinal flow in carotid occlusion is a very rare finding and usually transient in nature (Biousse Am J Ophthalmol 1998). To my experience, in these rare cases, there is a strongly reduced, but still detectable CRA Doppler signal. Why would you assume a hemodynamic and not embolic cause of SSN CRAO (for example: acute occlusion of a midgrade ICA stenosis and embolisation of a blood clot during the process of occlusion).

Line 187: What do you mean by "proximal disturbance of the ophthalmic artery due to atherosclerotic disease"? Is this an ultrasound diagnosis (which I can hardly imagine): If so, it should be explained in the results section and highlighted by a figure).

Author Response

Response to Reviewer 2 Comments

Dear Reviewer,

We appreciate all the reviewer's comments and suggestions.  Attached is a point-by-point response to the reviewer's comments, as well as a copy of our revised manuscript based on the response to those comments. We will be happy to provide additional information upon request or to modify the text further if still necessary.

Point 1: This study evaluated orbital ultrasound performed within 24h after ophthalmologically proven CRAO. An important finding is the confirmation of the ophthalmological diagnosis in all cases, which highlights the diagnostic strength of ultrasound in CRAO, thus reproducing literature data (Nedelmann et al Stroke 2015).

Response 1:  see comment “Point 1. Response to Reviewer 2”.

Reference “Nedelmann et al Stroke 2015” has been added as [20]

Page 7, line 174: “It is noteworthy that, these data replicate the findings described in the previous studies.” has been added.

Point 2: The results and discussion section are difficult to read and needs revision.

Response 2: see comments “Point 2. Response to Reviewer 2”.

We have substantially modified the text to make it more understandable, including introduction, results, discussion and slightly conclusions. 

The previous discussion was 950 words and is now 770 words long.

Point 3: There is a major issue that remains undiscussed. It is towards the nature of the spot sign. We do not know exactly its composition. But all circumstances are towards calcified or cholesterol particles and against pure blood clot: its bright reflex; it never disappears in follow-up or after thrombolysis: even after successful thrombolysis, it remains. That is clear evidence against atrial fibrillation as embolic source in SSP CRAO. This should be discussed more thoroughly. Respective literature should be cited and discussed (for example Altmann et al, J Neuroimaging 2015; Nedelmann et al Stroke 2015). In the context of this assumption, the finding of a strong association of heart valve calcification and SSP CRAO is the most interesting aspect of this study. It may well explain, why spot signs never disappear in follow-up (in contrast to most embolic spot sign negative CRAO that eventually recanalise spontaneously). To my knowledge, this finding has not been described previously.

Response 3: see comments “Point 3. Response to Reviewer 2”.

Reference “Altmann et al, J Neuroimaging 2015” has been added as [21]

We have discussed in depth the nature of the spot sign. Furthermore, we have added additional information on this topic: "Five (71,42%) of the 7 cases of major cardioembolic etiology (AF) in PSS group also had a minor cardioembolic source (left heart valves calcifications)”. This added information would now support the idea suggested by the reviewer that the emboli producing the spot sign would have a relevant calcium and/or cholesterol component.

In addition, we have modified the abstract, results, discussion, and conclusions to emphasize and deepen upon the potential nature of the spot sign. See comments “Point 3. Response to Reviewer 2”.

Point 4: Line 119: A stenosis >/= 50% is not high grade, which would be >/= 70%. This should be corrected. Please state, how many of your patients with ICA stenosis had a retrograde ophthalmic artery (a retrograde flow would effectively preclude embolism from stenosis to CRA

Response 4: see comments “Point 4. Response to Reviewer 2”.

High grade stenosis >/= 50% has been changed to “significant ICA stenosis >/= 50%”, which is according to TOAST classification for the cause of ischemic strokes, in: table 3, line 123, line 130, line 152, line 153, line 163, line 165, line 224 and line 226.

How many of our patients with ICA stenosis had a retrograde ophthalmic artery? à We did not detect any cases of retrograde ophthalmic artery.

Point 5: Line 147 and Table 5: I completely do not understand the assumption of hemodynamic CRAO in the stated context. Visual compromise due to low retinal flow in carotid occlusion is a very rare finding and usually transient in nature (Biousse Am J Ophthalmol 1998). To my experience, in these rare cases, there is a strongly reduced, but still detectable CRA Doppler signal. Why would you assume a hemodynamic and not embolic cause of SSN CRAO (for example: acute occlusion of a midgrade ICA stenosis and embolisation of a blood clot during the process of occlusion).

Response 5: see comments “Point 5. Response to Reviewer 2”.

In Table 5 and lines 149-151 we wanted to show the data from the point of view of the pathogenic mechanism and not the etiological mechanism. Table 5, lines 149-151 and some points relating to the results of the CRAO mechanism have been deleted to simplify the text and avoid confusing interpretations.

Regarding the hemodynamic mechanism of retinal ischemia, we only postulate it for cases with ipsilateral carotid artery occlusion. We have added that point to the discussion “We postulate a hemodynamic mechanism of retinal ischemia for these cases given that we found no flow signal in the CRA in all the 4 patients (in two of them there was even no flow in the ophthalmic artery). This hemodynamic mechanism of ischemia would be similar to that accepted for the so-called "ocular ischemic syndrome", where a high-grade stenosis or occlusion of the ICA, associated with poor collaterals of the external carotid branches, causes ischemia in the retina with a presumed hemodynamic pathophysiology”. See “Point 5. Response to Reviewer 2”.

We hypothesize a pathogenic mechanism of ischemia similar to that accepted for the so-called "ocular ischemic syndrome", where a high-grade stenosis or occlusion of the ICA, associated with poor collaterals of the external carotid branches, causes ischemia in the retina with a presumed hemodynamic pathophysiology. The absence of cases with retrograde flow in the ophthalmic artery supports the hypothesis of poor collateral supply in our patients. Although in both cases embolization of a clot could have occurred during the process of carotid occlusion, we think this is much more unlikely than in cases of significant ICA stenosis (>/= 50%) and, in fact, if we assume that the presence of a spot sign makes it easier to think of an embolic pathophysiology, precisely the 4 cases mentioned did not have a spot sign. 

Point 6: Line 187: What do you mean by "proximal disturbance” of the ophthalmic artery due to atherosclerotic disease"? Is this an ultrasound diagnosis (which I can hardly imagine): If so, it should be explained in the results section and highlighted by a figure).

Response 6: see comments “Point 6. Response to Reviewer 2”.

“proximal disturbance of the ophthalmic artery” is a hypothesis to explain the absence of flow in CRA and spot sign detection in NSS group with a ICA occlusion or GCA.

Of course, we have not been able to see an atherosclerotic disease in the ophthalmic artery by ultrasound.

We agree that the term “proximal disturbance” can lead to confusion, so we have changed that paragraph to improve comprehension: “only in the NSS group did we find cases of ipsilateral ICA occlusion. In our series there were only 4 potential cases: an ICA occlusion due to cardiac embolism and three ICA occlusions due to large vessel atheromatous disease. We postulate a hemodynamic mechanism of retinal ischemia for these cases given that we found no flow signal in the CRA in all the 4 patients (in two of them there was even no flow in the ophthalmic artery). This hemodynamic mechanism of ischemia would be similar to that accepted for the so-called "ocular ischemic syndrome", where a high-grade stenosis or occlusion of the ICA, associated with poor collaterals of the external carotid branches, causes ischemia in the retina with a presumed hemodynamic pathophysiology. “

We hope that these changes will provide a satisfactory response to the reviewer's comments.

We look forward to hearing from you soon. Thank you for your time and consideration.
